# Relationship between Cyberbullying, Motivation and Learning Strategies, Academic Performance, and the Ability to Adapt to University

**DOI:** 10.3390/ijerph182010646

**Published:** 2021-10-11

**Authors:** David Aparisi, Beatriz Delgado, Rosa M. Bo, María Carmen Martínez-Monteagudo

**Affiliations:** 1Department of Developmental Psychology and Didactic, Faculty of Education, University of Alicante, 03080 Alicante, Spain; david.aparisi@ua.es (D.A.); maricarmen.martinez@ua.es (M.C.M.-M.); 2Department of Research and Diagnostic Methods in Education, Faculty of Philosophy and Educational Sciences, University of Valencia, 46010 Valencia, Spain; rosa.m.bo@uv.es

**Keywords:** cyberbullying, academic motivation, learning strategies, ability to adapt to university, academic performance

## Abstract

Cyberbullying has become a frequent relational problem among young people, which has made it necessary to evaluate and prevent it in the university setting. The aim of this study is to examine the relationship between cyberbullying, motivation and learning strategies, the ability to adapt to university, and academic performance. A sample of 1368 Spanish university students (64% female) was administered a battery consisting of the European Bullying Intervention Project Questionnaire, the Learning and Study Strategies Inventory Short version, and the Student Adaptation to College Questionnaire, with their academic performance also being studied. The results found that the victimized bullies have greater difficulties in their organization and planning for study and exams, have fewer control and consolidation strategies, and are less able to adapt to university. Logistic regression analyses show that the greater the difficulties in organization and planning, and the greater the difficulties experienced in exams, the greater the probability of a person being a victim and a victimized bully. In addition, students are less likely to be victims, bullies, and victimized bullies as their ability to adapt to university increases. The findings have been discussed and it has been noted that there is a need to address academic adjustment and the ability to adapt to the university environment as a preventive measure for cyberbullying in university students.

## 1. Introduction

In recent decades, we have witnessed an unprecedented technological revolution that has allowed us to become more connected and expand our social networks worldwide. However, these technological and digital advances have not come without their risks, which include cyberbullying [1]. Cyberbullying is defined as the use of information and communication technologies (ICTs) by an individual or group that has the intention of harming others who cannot defend themselves. This is done deliberately, and in a repetitive and hostile manner [2]. Cyberbullying can be carried out using email, blogs, chats, messages, web pages, online games, and social networks, among other methods. Regarding the participative roles, while there are different classifications [3], we highlight that the classification composing of bullies, victims, victimized bullies, and uninvolved people or observers is the most sparing. The number of cyberbullying incidences is alarming among young people. Prevalence data in Spanish adolescents indicate that 8.8% are cybervictims, 3.1% are cyberbullies, and 4.9% are high-frequency cybervictim-bullies [4]. In addition, cyberbullying can occur at different ages and in any gender. However, it usually starts during adolescence and there are usually more female victims and more male bullies [5,6], and cyberbullying can be related to physical, cultural, racial, and even religious prejudices [7]. At university, this problem becomes chronic since, as certain studies highlight, when individuals in high school have been bullies or cybervictims, they tend to develop these same roles at university [8,9].

The consequences of cyberbullying are very serious for the victim at multiple levels. In this sense, victims show higher rates of anxiety [10], depression [11], permanent behavioral changes, such as resignation and social isolation, a predominance of obsessive traits with hostile attitude and hypersensitivity, and feelings of learned helplessness and low self-esteem [12]. At the academic level, they are likely to present more attention and concentration problems, have poor success in tasks, low academic performance, as well as seem to appear to be unmotivated or disinterested students [13] who refuse to go to school [14]. Thus, some studies have assessed the importance of cognitive-motivational and academic adjustment in the development of the problem of cyberbullying and cybervictimization in adolescents and youth. These studies have been detailed below.

### 1.1. Cyberbullying, Academic Motivation, and Learning Strategies

Most students are usually involved in their studies for different reasons, such as the desire to learn more and improve their skills, to demonstrate their ability, or to protect their self-image by seeking positive appraisals from others [15]. In this sense, research notes that there is a relationship between motivation, positive self-concept, and better academic performance [16]. Regarding the relationship between academic motivation and cyberbullying, a study examined the relationship between self-concept and academic goals with being a victim, bully, or being uninvolved in 548 Spanish students between 10–12 years old [17]. Logistic regression analyses showed that social self-concept and learning goals were protective factors for all three roles, with academic self-concept and achievement goals being protective variables for cybervictimization, and motivational orientation toward social reinforcement being a risk factor for perpetrating cyberbullying behaviors. Another cross-cultural study conducted with a sample of 3830 Spanish and Colombian adolescents found a negative relationship between normative adjustment and cyberbullying and a direct relationship between social motivation with cyberaggression. Cybervictimization was explained by prosocial behaviors and avoidance goals, and there was an inverse relationship with perceived social self-efficacy, developmental goals, and social and normative adjustment [18]. Regarding university populations, a research project conducted with 864 Spanish students, which intended to analyze the perception and experience of cyberbullying in a group of young university students in their secondary education stage, concluded that most of the young people who had suffered cyberbullying recognized that they did not feel motivated enough to attend classes at the time, nor properly develop their studies [19]. Motivation has been widely related to learning strategies as they refer to all cognitions, behaviors, beliefs, and emotions that facilitate the acquisition, understanding, and subsequent transfer of new knowledge or a new skill [20]. The relationship between cyberbullying and learning strategies has been scarcely analyzed by the scientific community. In this sense, there are studies that relate collaborative and cooperative learning strategies to cyberbullying. An investigation conducted with 360 Chinese and Pakistani students found cyberbullying to be a moderator that decreases the positive relationship between collaborative learning and student achievement [21]. In this regard, previous research findings indicate that students who experience cyberbullying feel less focused on their studies [22]. Another investigation conducted with 1052 Israeli higher education students showed, through logistic regression analysis, a positive relationship between learning difficulties and cybervictimization [23]. Moreover, the study conducted by Al-Rahmi et al. [24] on a sample of 538 Malaysian university students demonstrated the negative relationship between cyberbullying, academic performance, and collaborative learning strategies. These study results underscore the implication of cyberbullying on the maladaptive use of strategies in studying, with there being lower collaborative strategies and concentration among students who have been cyberbullied. However, it remains to be clarified whether study and learning strategies, such as planning and organization, control and the prioritization of knowledge, and difficulties during test-taking may lead to there being an increased risk of cyberbullying in all three roles.

### 1.2. Cyberbullying and the Ability to Adapt to University

When starting university, students may face situations that impact their performance. Effects on performance may be due to several psychological and social causes that influence academic success [25]. As such, one of the main problems affecting higher education internationally is the increased drop-out rate from university studies [26]. Regarding the relationship between cyberbullying and a student’s ability to adapt to university, reviews on the subject systematically note that cyberbullying increases during secondary education but decreases in later teenage years [27]. However, the increase in cyberbullying cases in recent years has meant more interest in the problem in the university setting [28]. There are very few studies that analyze the specific relationship between cyberbullying and a student’s ability to adapt to university. Most studies have focused on analyzing the effects of cyberbullying in the university population. A recent investigation [29], using a sample consisting of 1653 Spanish first-year university students, analyzed whether those university students who were victims of bullying (traditional bullying and cyberbullying) had a higher drop-out rate. The results showed that those students who were victims, compared to those who were not, considered, to a greater extent, leaving their studies, with variables being related to social integration, such as receiving support from friends and professors also having a moderating effect. The study by [30] analyzed the predictive capacity of certain emotional problems (anxiety, depression, and stress) and the ability to adapt to university with respect to cyberbullying in 1282 Spanish university students. The results showed that high levels of depression and stress increased the probability of being a victim of cyberbullying, while high levels of depression increased the probability of being a cyberbully. Similarly, the students’ personal, emotional, and social adaptation decreased the likelihood of being a victim of cyberbullying. Taking this evidence into account, it should be clarified whether there is a relationship between the ability to adapt to university and all roles involved in cyberbullying, including bullies or victimized bullies.

### 1.3. Cyberbullying and Academic Performance

According to [31], academic performance is a construct that is able to take on quantitative and qualitative values, through which it has a closeness to the evidence and dimension of the profile of skills, knowledge, attitudes, and values that the student develops in the teaching-learning process. The main indicator to measure academic performance is the average of the grades obtained by the student in a given school period [32]. Regarding the relationship between cyberbullying and academic performance, the literature review shows inconclusive results. While most studies have concluded that there is a negative association between cyberbullying and academic performance [33,34,35], others note that the relationship is not significant and that the impact of traditional social-type bullying on academic performance is greater [36]. Thus, the study developed by [33], with 3451 Spanish students aged 12 to 19 years, concluded that young people with lower emotional intelligence were more likely to suffer cyberbullying and could experience negative repercussions on their school success with poor academic performance. Okumu et al. [34], with a representative sample of U.S. university students, concluded that cyberbullying is associated with poor academic performance. Similar results were found by [35], who, through a study with 413 American students aged 17 to 19 years, found that young people who had been cyberbullied showed greater academic difficulties and poorer academic performance, although this negative effect was buffered by perceived parental social support. Therefore, this relationship should be analyzed with a view to clarifying the need to develop actions in academic matters with students involved (victims, bullies, victimized bullies) in cases of cyberbullying.

### 1.4. The Present Study

Although previous empirical evidence has highlighted the relationship between cyberbullying and several educational variables such as academic motivation and learning strategies [16,18,19], the ability to adapt to university [29,30], and academic performance [34,35,36], there is a lack of studies that specifically examine the relationship between cyberbullying and such educational variables in university students. Therefore, the aim of this research is twofold: (1) to study the differences between the different roles involved in cyberbullying (victims, bullies, victimized bullies) and those uninvolved in it with respect to motivation and learning strategies, the ability to adapt to university, and academic performance; and (2) to analyze the predictive capacity of academic motivation and learning strategies, the ability to adapt to university, and academic performance on cyberbullying in its three main roles in a sample of Spanish university students. From the review of the previous research, it was expected to find differences in the roles involved in cyberbullying attending to different educational variables. More specifically, it was expected that uninvolved students present higher scores in motivation and learning strategies than students involved in cyberbullying cases (Hypothesis 1). Regarding the ability to adapt to university, it was expected that victims, bullies, and victimized bullies present lower levels compared to those not involved (Hypothesis 2). Regarding the role of the victim, these students are expected to present lower scores in academic performance, as well as low grades, explaining their involvement in cyberbullying (Hypothesis 3). Finally, motivation and learning strategies, the ability to adapt to university, and academic performance are expected to be significant predictors of the different roles involved in cyberbullying (Hypothesis 4).

## 2. Materials and Methods

### 2.1. Participants

The reference population was university students in the Valencian Community (Spain). Students were randomly selected from two public universities in the provinces of Alicante and Valencia. Once the universities were selected, fourteen classes were randomly selected from each center. Due to the random sampling method, the socioeconomic status and ethnic compositions of the overall sample are assumed to be representative of the community. Of the 1404 students recruited (740 from the University of Valencia and 664 from the University of Alicante), 36 were eliminated due to omissions or errors in the tests. Therefore, a total of 1368 university students (494 male; 36% and 874 female; 64%) participated in the research in the following academic years: 1st year (45%), 2nd year (21.9%), 3rd year (12.1%), and 4th year (20.9%). The mean age of the participants was between 18 and 49 years (*M* = 21.34; *SD* = 4.45). By means of the Chi-square test, used to analyze the homogeneity of the frequency distribution, it was found that there were no statistically significant differences between the sex of the participant and the course groups (*χ*^2^ = 18.44; *p* > 0.05).

### 2.2. Instruments

#### 2.2.1. European Bullying Intervention Project Questionnaire (EBIPQ)

The EBIPQ [37,38] is a measure widely used in European research projects to assess the frequency and intensity of cyberbullying in adolescents and young people. It is a scale consisting of 22 Likert-type items with five response options, with a scoring system between 0 (*never*) and 4 (*always*). It consists of two dimensions: cybervictimization (“someone has threatened me through internet messages”) and cyberaggression (“I have spread rumors about someone on the internet”). For both dimensions, the items refer to actions such as swearing, excluding the victim, spreading rumors, impersonating, and so forth, with all actions happening through electronic media and within a time interval of the previous two months. The scale has evidence of being reliable in the scores of its subscales [38]. In the present study, adequate reliability indices were obtained for the subscale of cybervictimization (α = 0.80) and cyberaggression (α = 0.88).

#### 2.2.2. Learning and Study Strategies Inventory Short Version (LASSI-S)

The LASSI-S [39,40] is a questionnaire that evaluates a student’s motivation and the use of different learning strategies during study. The short version includes a total of 21 items, using a Likert scale with values ranging from 1 to 5, from “*Does not describe me at all*” to “*Describes me a lot*”. The questionnaire is composed of six subscales: Organization and planning (“I find it difficult to organize and plan how I study and stick to it”), Test performance skills (“I have difficulty understanding test questions”), Skills for prioritizing information (“I have poor ability to summarize what I read or hear”), Learning resources (“I make diagrams or graphs to summarize the contents of a subject”), Control and consolidation strategies (“after class, I reread my notes to better understand the information”), and Motivation (“even when what I have to study is boring, I manage to keep working until I finish”). The reliability and validity indicators obtained by the original authors were satisfactory [39]. The range of reliability indicators (Cronbach’s alpha) of the subscale scores ranged from 0.70 (Examination Difficulties) to 0.83 (Learning Resources).

#### 2.2.3. Student Adaptation to College Questionnaire (SACQ)

The SACQ [41] is a 50-item self-report designed to measure the students’ ability to adapt to the university environment. This questionnaire presents full-scale test scores and four subscales: Social, Academic, Emotional, and Personal Adjustment. Participants who take this questionnaire are evaluated on a 5-point Likert-type scale ranging from 1 (“*Does not fit me at all*”) to 5 (“*Fits me perfectly*”). The test measures a student’s success in coping with various educational demands in terms of their university experience, efficacy in coping with interpersonal social demands at university, feelings about their physical and psychological state, and an assessment of the overall university experience (“I am satisfied with my decision to attend college”). Baker and Siryk [41] achieved sufficient reliability for each of the subscales and for the satisfactory overall score (α > 0.80). In this investigation, an overall score drawn from the questionnaire items was used, using an adequate scale reliability indicator (α = 0.82).

#### 2.2.4. Academic Performance

For the evaluation of academic performance, the number of failed subjects, and the average grades obtained across the subjects of an academic year were taken into account, following the numerical weighting of 1 (Fail = scores between 0–4.99), 2 (Pass = scores between 5–6.99), 3 (Merit scores between 7–8.99), and 4 (Outstanding = scores between 9–10).

### 2.3. Procedure

First, once the centers had been selected, a meeting was held with the management teams of the faculties in order to explain the objectives of the research work and the evaluation instruments to be used, to request their permission, and encourage their collaboration in the research. The questionnaires were completed voluntarily and were done collectively during a class session, ensuring the anonymity of the participants by means of identification numbers on the answer sheets. The researchers were present during the completion of the tests to clarify possible doubts and verify that the correct administration had been done. Emphasis was placed on the total completion of the tests, with an average time of approximately 35 minutes being used to do them. The research was approved by the ethics committee of the study center and complied with the postulates of the Helsinki declaration for human research.

### 2.4. Statistical Analyses

Firstly, the sample was classified into bullies, victims, victimized bullies, and those uninvolved. Taking into account the scores on the ECIP-Q questionnaire, students were classified as follows: cybervictims were those students who obtained scores equal to or higher than 1 (Yes, once or twice) in any of the items on cybervictimization, and whose scores were equal to 0 (Never) in all the items on cyberaggression; cyberbullies were those with scores equal to or higher than 1 (Yes, once or twice) in any of the items on cyberaggression and whose scores were equal to or lower than 0 (Never) in all the items on cybervictimization; and victimized cyberbullies were those with scores in any of the items on cyberaggression and cybervictimization being equal to or higher than 1 (Yes, once or twice). Secondly, the difference of means (Fisher’s *F* statistic) was calculated for the variables of motivation, learning strategies, ability to adapt to university, and academic performance, and statistically significant differences were analyzed using a Bonferroni post hoc test. In addition, in order to identify the magnitude of the differences found between the groups, the *d* index proposed by Cohen [42] was calculated. Its interpretation is as follows: a small effect size is found at values of 0.20 ≤ *d* ≤ 0.49, moderate between 0.50 ≤ *d* ≤ 0.79, and large at values *d* ≥ 0.80. Finally, to evaluate the predictive capacity of the different educational variables on cyberbullying (bully, victim, and victimized bully), a logistic regression analysis was carried out through the forward stepwise procedure based on the Wald statistic. The quantification of the probability of occurrence of an event (e.g., being a victim of cyberbullying) was performed through the Odds Ratio (*OR*), whose interpretation is as follows: if the *OR* is greater than one, for example three, for each occurrence of the event in the presence of the independent variable, it will occur three times if this variable is present. On the other hand, if the *OR* is less than one, for example 0.5, the probability of the event occurring in the absence of the independent variable will be greater than in its presence. The fit of the proposed models was evaluated on the basis of Nagelkerke’s R^2^ and the percentage of cases correctly classified by the model. All analyses were carried out using SPSS version 26.0 statistical software.

## 3. Results

### 3.1. Differences in Motivation and Learning Strategies, the Ability to Adapt to University, and Academic Performance in Victims, Bullies, Victimized Bullies, and Those Uninvolved in Cyberbullying

In the study, 15.5% of respondents met the criteria for being pure cybervictims, 7.5% were pure cyberbullies, 60.7% were identified as victimized bullies, and 16.2% were categorized as uninvolved. The results of the ANOVA test (see Table 1) indicated the existence of statistically significant differences in three learning strategies (difficulties in organization and planning, in taking exams, and in performing control and consolidation strategies) and in the ability to adapt to university (*p* < 0.001).

Regarding learning strategies, the post hoc contrasts indicated that the victimized bully students scored significantly higher in terms of the difficulties in organization and planning in the study with respect to the uninvolved students (*d* = 0.47) and pure victim students (*d* = 0.34). In addition, victimized bully students presented greater difficulties during exams than the victims (*d* = 0.80), the pure bullies (*d* = 0.37), and the uninvolved (*d* = 0.85) students. On the other hand, pure bullies presented greater difficulties in exams than victims and uninvolved students, with the size of these differences being large (*d* = 0.88 and *d* = 0.90, respectively). Furthermore, victimized bullies and pure bullies obtained significantly lower scores in control and consolidation strategies than the victims and uninvolved students, with the effect size of these differences being low (*d* < 0.28). With respect to the dimension of the ability to adapt to university, victimized bully students scored significantly lower than uninvolved students, with the effect size of these differences being low (*d* = 0.21). As for the differences in academic performance, lower scores in academic performance and more cases of students failing were found in those involved in cyberbullying compared to those not involved, although no significant differences were found for these differences (*p* > 0.05).

### 3.2. Prediction of the Role of the Victim through Academic Variables

From the logistic regression analysis, it was possible to create two predictive models for being a victim of cyberbullying from learning strategies and the ability to adapt to university (Table 2), each correctly classifying 76.2% of the cases (χ^2^ = 150.69; *p* = 0.001) and 76.3% (χ^2^ = 7.09; *p* = 0.001), respectively. The fit value (Nagelkerke’s R^2^) was 0.16 for the first model and 0.01 for the second model. The Odds Ratios (*OR*) indicated that students are 6% more likely to be victims of cyberbullying, with respect to the uninvolved group, as their score on planning and organization difficulties scale increases by one unit and are 219% more likely to be victims as their score on exam difficulties scale increases by one unit. With respect to the ability to adapt to university, the *OR* indicates that students are 4% less likely to be victims of cyberbullying as their score regarding their ability to adapt to university increases by one unit. These results indicate that adaptation to university and educational counseling are important factors to take into account in the prevention of cyberbullying.

### 3.3. Prediction of the Role of the Bully through Academic Variables

From the logistic regression analysis, it was possible to create a predictive model of being a cyberbully from the ability to adapt to university (Table 3), correctly classifying 68.2% of the cases (χ^2^ = 9.83; *p* = 0.00). The fit value (Nagelkerke’s R^2^) was 0.1. The *OR* indicates that students are 5% less likely to be cyberbullies (relative to the uninvolved group) as their ability to adapt to university score increases by one unit. In this sense, the feeling of relevance and well-being in the university can prevent the appearance of cyberbullying.

### 3.4. Prediction of the Role of the Victimized Bully through Academic Variables

From the logistic regression analysis, two predictive models of being a cyberbully were created from learning strategies and the ability to adapt to university (Table 4), correctly classifying 72.2% of the cases (*χ*^2^ = 422.09; *p* = 0.00) and 60.8% of the cases (*χ*^2^ = 13.22; *p* = 0.00), respectively. The fit values (Nagelkerke’s R^2^) for the models were 0.36 and 0.01, respectively. The Odds Ratios (*OR*) indicate that students are 8% more likely to be victimized bullies, with respect to the group of those not involved, as their score regarding difficulties in planning and organization of studies increases by one unit, and are 693% more likely as their score regarding difficulties in exams increases by one unit. In addition, students are 5% less likely to be victimized bullies as their score regarding their ability to adapt to university increases by one unit. In this sense, we can observe how educational variables that represent a difficulty for students can contribute to their becoming victimized bullies.

## 4. Discussion

The aim of this study was to analyze the relationship between cyberbullying, academic motivation, learning strategies, the ability to adapt to university, and academic performance in a sample of Spanish university students.

Unlike previous research, this study analyzes the importance of the ability to adapt to university based on the evaluation of the different roles of cyberbullying. In addition, and also unlike previous studies, this research has contemplated such relationship taking into account the analysis of effect sizes to determine the magnitude of the differences found, which was recommended by different authors [42,43]. In addition, this study establishes the predictive analysis of cyberbullying in its main roles, relying on an instrument that has been validated in a large European sample and which collects the defining characteristics of cyberbullying [38].

Considering the results obtained, in the case of victimized bullies and bullies, it was possible to confirm the first hypothesis since they presented greater difficulties in organizing and planning their studies, as well as in adequately performing in exams, and developed less control and consolidation strategies than uninvolved students. Such results are congruent with previous research that found lower uses of collaborative learning strategies [21,24] and lower concentration in students involved in cyberbullying [22,44]. This evidence underlines the importance that cyberbullying has on the use of study strategies and their efficacy in academics, especially among students who are simultaneously victims and bullies and among those who are pure cyberbullies. In this sense, the learning process in these students may be affected by maladjusted strategies when planning for study and in test performance. It may also be affected by lower review and comprehension strategies and more concentration problems [22,44]. These difficulties in studying can lead students to give low value to their studies and have greater rejection towards school [14,45].

Regarding the ability to adapt to university, it was expected that victims, bullies, and victimized bullies would present lower levels compared to those not involved in cyberbullying (Hypothesis 2). The results partially confirmed the second hypothesis as victimized bullies scored significantly lower than uninvolved students in their ability to adapt to university. This result suggests that being simultaneously a victim and a bully is related to lower coping skills in the face of novel situations such as starting university and in the quality of established social relationships [45,46]. Victimized bully students feel less attached and satisfied with university, which may lead them to consider early drop-out [29]. This imbalance may be due to the social interaction difficulties they develop [47], as well as their lower ability to adequately manage stress, manage conflict, solve problems in the face of novel situations, and their low self-control. Although the results are consistent with previous studies that point to a lower ability to adapt to university [30], these findings should be viewed with caution because of the low magnitude in the effect size of the differences. It is important to consider that, although there is a lower ability to adapt to university in victimized bully students, such difference is small and, therefore, its theoretical-practical relevance should continue to be examined in future studies.

Regarding academic performance, the third hypothesis could not be confirmed since there were no statistically significant differences between the different groups involved (victims, bullies, and victimized bullies) and those not involved in cyberbullying. This result differs from what has been found in previous research referring to a relationship between low academic performance and being a victim of cyberbullying [33,34,35]. However, this coincides with what was found by Torres et al. [36] who confirmed that the explanatory weight of cyberbullying for academic performance was not significant when assessing other types of bullying such as social bullying. Furthermore, this finding could account for the importance of other personal and social variables that explain the phenomenon of cyberbullying, such as emotional intelligence, self-concept, and perceived stress [30,48], as well as problematic internet use [49] or attitudes towards bullying behaviors and perceived social support [47].

Finally, from the results obtained in the logistic regression analyses, it was possible to partially confirm the fourth hypothesis, as students were more likely to be victims and victimized bullies of cyberbullying as their scores on the scales of difficulties in exams (*OR* = 3.19–7.93) and in organization and planning of studies (*OR* = 1.06–1.08) increased and were less likely as their scores regarding their ability to adapt to university increased (*OR*= 0.95–0.96). On the other hand, students presented less likelihood of being cyberbullies as their score regarding their ability to adapt to university was higher (*OR* = 0.95). These findings are consistent with previous studies analyzing the academic and study profile of cyberbullying victims [21,22,24,44]. These findings suggest that difficulties in the use of study and learning strategies are also able to be extrapolated, with this being the case, to a greater degree, in students who are simultaneously victims and bullies. Therefore, correct preparation before evaluations and an improvement in concentration, as well as in organization and planning in studies, can be actions to reduce the risk of cyberbullying among university students. Furthermore, adjustment within the university environment is especially relevant for the explanation of this problem, as previous studies have highlighted [29,30] since it is an explanatory variable of the three roles involved in cyberbullying. Young people who feel more comfortable, safe, and satisfied at university show less risk of developing online bullying problems. Therefore, it would be advisable for higher education institutions to prioritize the establishment of programs for the improvement of the ability to adapt and the transition to university with actions using the Tutorial Action Plan, especially among dissatisfied young people, with greater difficulties in adaptation or who receive less social support from their environment.

This research has some limitations. First, although the sampling method used guarantees the representativeness of the sample recruited, the results found in this study cannot be generalized to students at other educational levels. Future research should confirm whether the results found in university students hold true at other educational levels. Furthermore, it would be advisable for future works to use longitudinal designs in order to provide more conclusive data regarding the causal relationships between these variables. We should also take into account the modulating effect of other educational (academic self-concept), personal (self-esteem, emotional intelligence), social, and family variables that may mediate the relationship between cyberbullying and the study variables. Finally, this research aims to understand the predictive capacity of motivation, learning strategies, the ability to adapt to university, and academic performance on the different roles of cyberbullying and not the other way around (the predictive capacity of the different roles of cyberbullying on motivation, learning strategies, the ability to adapt to university, and academic performance). Although it is logical to think that there is a reciprocal effect, future research could analyze this question by developing structural equation models to test which hypothesis is the most tenable.

On a practical level, the results of this research, firstly, support the effectiveness of programs aimed at enhancing students’ ability to adapt to university, as it has been found to prevent the risk of cyberbullying in the context of higher education. Secondly, the research has focused on the protective and vulnerability factors of victims, bullies, and victimized bullies. In this regard, one of the variables that is negatively associated with cybervictimization is an incorrect use of certain learning strategies in the process of organization and planning studies, as well as during the process of taking exams. Therefore, it is essential to establish and improve these strategies in university students in order to develop more effective preventive programs [50]. Finally, social support from family, teachers, and friends could reduce negative psychosocial symptoms while increasing the well-being of students involved in cyberbullying [51,52]. Therefore, it would be beneficial for future lines of analysis to include school and family factors as moderating variables in this analysis with a firm purpose of preventive action for the future.

## 5. Conclusions

In conclusion, this research confirms the existence of statistically significant differences in learning strategies and the ability to adapt to university, according to the role involved in cyberbullying. In the case of academic performance, the differences were not statistically significant for the roles analyzed. Regression analyses showed that learning strategies and the ability to adapt to university were statistically significant predictors of the different roles of cyberbullying, since students with high scores for difficulties in exams and for planning and organizing their studies were more likely to be victims and victimized bullies; and students with high scores for the ability to adapt to university were less likely to be victims, bullies, and victimized bullies.

## Figures and Tables

**Table 1 ijerph-18-10646-t001:** Differences in means and standard deviations of personality traits and aggressiveness between victims and pure bullies, victimized bullies, and uninvolved students.

	Victim	Bully	Victimized Bully	Uninvolved	Statistical Significance	
	*M (SD)*	*M (SD)*	*M (SD)*	*M (SD)*	*F*	*p*
Difficulty in organization and planning	17.91 (4.82)	18.28 (5.33)	19.52 (4.73)	17.24 (5.08)	16.40	0.001
Difficulty in exams	3.08 (0.43)	3.56 (0.70)	4.02 (1.30)	3.01 (0.58)	93.24	0.001
Difficulty in prioritizing information	6.67 (2.28)	6.37 (2.48)	6.79 (2.26)	6.43 (2.07)	2.18	n.s.
Motivation	11.20 (2.43)	11.08 (2.53)	10.89 (2.59)	11.36 (2.58)	2.33	n.s.
Learning resources	6.98 (2.31)	6.38 (2.65)	6.61 (2.42)	6.75 (2.66)	1.84	n.s.
Control and consolidation	7.70 (2.72)	7.13 (2.49)	7.32 (2.57)	7.78 (2.67)	3.02	0.02
Ability to adapt to university	23.75 (3.91)	23.75 (3.87)	22.96 (4.21)	23.84 (4.10)	4.40	0.001
Academic performance	3.84 (0.52)	3.77 (0.59)	3.81 (0.56)	3.77 (0.49)	0.87	n.s.
Number of subjects failed	1.35 (0.81)	1.31 (0.67)	1.32 (0.66)	1.26 (0.59)	0.67	n.s.

*Note*. n.s = not significative.

**Table 2 ijerph-18-10646-t002:** Results derived from binary logistic regression for the probability of being a victim of cyberbullying.

Predictive Variable	B	S.E.	Wald	*p*	*OR*	C.I. 95%
Difficulties in Exams	1.16	0.14	72.28	0.00	3.19	2.44–4.17
Difficulty in Organization and Planning	0.06	0.01	18.66	0.00	1.06	1.03–1.09
Constant	−3.87	0.51	58.03	0.00	0.02	
Ability to adapt to university	−0.04	0.02	6.92	0.00	0.96	0.93–0.99
Constant	2.15	0.38	31.72	0.00	8.55	

*Note*. B = coefficient; S.E. = standard error; *p* = probability; *OR* = odds ratio; C.I. = confidence interval at 95%.

**Table 3 ijerph-18-10646-t003:** Results derived from binary logistic regression for the probability of being a cyberbully.

Predictive Variable	B	S.E.	Wald	*p*	*OR*	C.I. 95%
Ability to adapt to university	−0.04	0.01	9.60	0.00	0.95	0.93–0.98
Constant	1.81	0.35	27.59	0.00	6.15	

*Note*. B = coefficient; S.E. = standard error; *p* = probability; *OR* = odds ratio; C.I. = confidence interval at 95%.

**Table 4 ijerph-18-10646-t004:** Results derived from binary logistic regression for the probability of being a cybervictimized bully.

Predictive Variable	B	S.E.	Wald	*p*	*OR*	C.I. 95%
Difficulty in Organization and Planning	0.08	0.01	31.70	0.00	1.08	1.05–1.11
Difficulties in Exams	2.07	0.15	190.56	0.00	7.93	5.90–10.63
Constant	−7.95	0.56	201.95	0.00	0.00	
Ability to adapt to university	−0.05	0.01	12.93	0.00	0.95	0.92–0.97
Constant	1.59	0.33	23.65	0.00		

*Note.* B = coefficient; S.E. = standard error; *p* = probability; *OR* = odds ratio; C.I. = confidence interval at 95%.

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
