# Peer review of "Relationship between Cyberbullying, Motivation and Learning Strategies, Academic Performance, and the Ability to Adapt to University"

_ijerph, 2021, doi:10.3390/ijerph182010646_

Round 1

Reviewer 1 Report

Dear authors. 

Firstly, I would like to tell you that in general, the manuscript you sent to the Journal is quite complete.

However, I need to highlight some aspect that should be improved:

- Firstly, there are several mistakes in the introduction part, related to the references... so the name of many authors is in the text... Please check it (Line 117, 129, 137, 142...). As you know, you may only use the number of the reference.

- As for the Results: The analisys you used were apropiated. Nevertheless the writting of the results (3.2; 3.3.; 3.4) it is could be confusing for the reader. I would suggest you to try to make it more clear and understandable. 

Author Response

Revisor 1

Dear authors. 

Firstly, I would like to tell you that in general, the manuscript you sent to the Journal is quite complete.

We are very grateful to the Reviewer for the positive evaluation of the work.

However, I need to highlight some aspect that should be improved:

- Firstly, there are several mistakes in the introduction part, related to the references... so the name of many authors is in the text... Please check it (Line 117, 129, 137, 142...). As you know, you may only use the number of the reference.

We appreciate the comment. We have corrected the indicated references in the text.

- As for the Results: The analisys you used were apropiated. Nevertheless the writting of the results (3.2; 3.3.; 3.4) it is could be confusing for the reader. I would suggest you to try to make it more clear and understandable. 

Thank you very much for your appreciation. We have added a brief explanation in each section to clarify the interpretation of the results for the reader.

Reviewer 2 Report

This manuscript examined the relationships among cyberbullying, motivation and learning strategies, academic performance, and the ability to adapt to university.  The topic and results of this manuscript are important for their implications on the prevention and intervention at the university level.  Overall, the manuscript is well-organized and written with appropriate literature review and research design.  There are three suggestions for the authors to revise:

  1. Page 5, 2.2.4 Academic performance: Please describe specifically how the weighting of 1, 2, 3, and 4 were classified by the authors.
  2. Page 5, 2.3 Procedure: Please describe how the centers were selected? What’s the participation rate for the centers and for students?
  3. Please describe more regarding the research participants’ demographic characteristics.

Author Response

Revisor 2

This manuscript examined the relationships among cyberbullying, motivation and learning strategies, academic performance, and the ability to adapt to university.  The topic and results of this manuscript are important for their implications on the prevention and intervention at the university level.  Overall, the manuscript is well-organized and written with appropriate literature review and research design. 

We are very grateful to the Reviewer for the positive evaluation of the work.

There are three suggestions for the authors to revise:

  1. Page 5, 2.2.4 Academic performance: Please describe specifically how the weighting of 1, 2, 3, and 4 were classified by the authors.

Thank you very much for your suggestion. Exact classification criteria have been added to the manuscript.

  1. Page 5, 2.3 Procedure: Please describe how the centers were selected? What’s the participation rate for the centers and for students?

Thank you very much for your suggestion. The explanation of the participants and the selection method in the participants section of the manuscript has been completed.

  1. Please describe more regarding the research participants’ demographic characteristics.

An explanation about the characteristics of the sample has been added.

Reviewer 3 Report

The paper as a whole is well structured and reports clear objectives of the research, has a broad reference to the literature and robustness of the satatistic analysis methodology adopted. However, two parts should be improved:
1 - The reference to literature often does not distinguish the geographic origin or age of the students involved in the researches commented on. This could lead to confuse the basic hypotheses when referring to experiments related to age groups very different from the target ones of the present work or in very different geographical contexts.
2 - The sample is not of a probabilistic type, at least from what can be deduced and due to the fact that the sampling design is not carried out. This fact should be better argued and explained how the interviewees were selected.

Author Response

Revisor 3

The paper as a whole is well structured and reports clear objectives of the research, has a broad reference to the literature and robustness of the satatistic analysis methodology adopted.

We are very grateful to the Reviewer for the positive evaluation of the work.

However, two parts should be improved:

1 - The reference to literature often does not distinguish the geographic origin or age of the students involved in the researches commented on. This could lead to confuse the basic hypotheses when referring to experiments related to age groups very different from the target ones of the present work or in very different geographical contexts.

We appreciate your comment. All age groups and ethnic origin of the cited studies have been added to the manuscript.

2 - The sample is not of a probabilistic type, at least from what can be deduced and due to the fact that the sampling design is not carried out. This fact should be better argued and explained how the interviewees were selected.

We appreciate your comment. The explanation of the participants and the selection method in the manuscript has been completed, being as follows:

The reference population was university students in the Valencian Community (Spain). Students were randomly selected from two public universities in the provinces of Alicante and Valencia. Once the universities were selected, fourteen classes were randomly selected from each center. Due to the random sampling method, the socioeconomic status and ethnic compositions of the overall sample are assumed to be representative of the community. Of the 1,404 students recruited (740 from the University of Valencia and 664 from the University of Alicante), 36 were eliminated due to omissions or errors in the tests. Therefore, a total of 1,368 university students (494 male; 36% and 874 female; 64%) participated in the research in the following academic years: 1st year (45%), 2nd year (21.9%), 3rd year (12.1%), and 4th year (20.9%). The mean age of the participants was between 18 and 49 years (M= 21.34; SD= 4.45). By means of the Chi-square test, used to analyze the homogeneity of the frequency distribution, it was found that there were no statistically significant differences between the sex of the participant and the course groups (χ2 = 18.44; p > .05).